# Grumpy Dogs Are Smart Learners—The Association between Dog–Owner Relationship and Dogs’ Performance in a Social Learning Task

**DOI:** 10.3390/ani11040961

**Published:** 2021-03-30

**Authors:** Péter Pongrácz, Gabriella Rieger, Kata Vékony

**Affiliations:** Department of Ethology, ELTE Eötvös Loránd University, 1053 Budapest, Hungary; riegergabriella@gmail.com (G.R.); kata.vekony.kami@gmail.com (K.V.)

**Keywords:** dog, social learning, detour test, dog–owner relationship

## Abstract

**Simple Summary:**

Dogs show considerable individual variability of success in social learning tasks. One of the factors associated with this variance is the dogs’ social rank, but no studies so far investigated the role of dog–owner relationship in this matter. We investigated how this relationship, with a focus on behaviour-problems, might affect social learning from the owner and strangers. We used a questionnaire and two behaviour tests to assess the dogs’ attitude towards their owner, and tested the dogs in a well-established detour reaching task without demonstration (individual problem-solving), with owner demonstration and with experimenter (unfamiliar human) demonstration. In general, dogs detoured faster if a human demonstrated the task. In case of unfamiliar demonstrator, dogs that scored higher on aggression and assertiveness-related traits learned better from the demonstrator, but also dogs who had low scores on possessiveness performed better in this condition. Traits relating to attention problems and activity did not affect performance, but these “overactive” dogs looked back at the owner less frequently during the individual problem-solving scenario. Our results indicate that dog–human relationship may have a complex association with various aspects of social interactions between the two species, including problem behaviours and social learning, too.

**Abstract:**

We investigated how dog–owner relationship–with a focus on possible behavioural problems–might associate with the individual variability in dogs’ social learning performance. Dog owners first completed a questionnaire about their relationship with their dogs (N = 98). Then, dogs were tested in a detour test: a control group without demonstration, a group where the owner demonstrated the task and another group where the experimenter demonstrated the task. Finally, the dogs participated in two behaviour tests measuring their tractability and possessiveness. The two principal components from the questionnaire (called “overactive” and “irritable”) did not show significant association with dogs’ detour performance in the control group. “irritable” dogs performed better in the unfamiliar demonstrator group. These more persistent, goal-oriented dogs also looked back less at their owners during the detour. In the individual problem-solving context, the factor “overactive” had a similar effect on looking back at the owner, suggesting that the items of this component primarily are not connected to the dog–human relationship. Our results indicate that dog–human relationship has an integral role in the complex social behaviour of dogs, which warrants for the need of further empirical testing of the associations between social dynamics in dogs and their relationship with humans, including problem behaviours.

## 1. Introduction

Various theories exist about the time, place [1,2] and main driving force of the domestication of dogs [3,4]. Although the ecological circumstances may vastly differ between the extant dog populations (from the feral/pariah dogs to working and family dogs [5]), there is an agreement between the scientists that the success of the dog as a species could be only guaranteed by its near-perfect adaptation to the given anthropogenic niche [6]. The main component of this adaptation (beyond the physiological and anatomical features [7,8]) manifests itself on the behavioural and socio-cognitive level [9]. Whether the human-compatible cognitive and behavioural traits are mostly based on directional selection [10] or ontogenetic and experience-related processes [11] is still debated, however, the main components of this socio-cognitive skillset of the dog have been already well outlined. Dogs are extremely easy to socialize to humans [12] and socialized dogs show the capacity for spontaneous (i.e., non-trained) bi-directional communication with humans (even with rudimentary levels of socialization, such as the pariah dogs [13]), attachment [14], rudimentary theory of mind (attributing knowledge to others [15], attributing intention [16]) and social learning [17]. It was found earlier that dogs prefer choosing a socially reinforced solution in a problem-solving task over the developing of their own trial-and-error methods. According to our earlier study [17], when dogs could observe at first a human demonstrator who detoured around an obstacle, they tended to choose the detour as a solution later, instead of using a shortcut (a door) across the obstacle. Thus, one can conclude that social learning is one of the main sources of behavioural synchronization between dogs and humans, where individual problem-solving and cooperation among conspecifics [18] was replaced by the preponderance of dog–human information transfer [19].

Considering the importance of the fundamental interspecific socio-cognitive capacities of dogs, it is assumed that these will show almost no, or only slight variance across the dog populations. As an indirect support for the aforementioned assumption, the various approaches to personality traits in dogs usually lack including such items as attachment, communication or social learning [20], although recently it was shown in connection to the Dognition.com project that the trait “communication” has a strong inherited background in dogs [21]. Animal personality refers to a variability among individuals that shows a temporal consistency and an across-context nature within the individual [22]. On the other hand, there are indications that anatomical (e.g., head shape) and functional (e.g., selection for cooperative or independent work tasks, [23]) and even life history factors (e.g., being a “shelter dog” [24]) may affect such behavioural skills as the following of human pointing gestures. The capacity for social learning (from human demonstrators) however, proved to be a rather stable feature in the tested dog populations so far, proving to be equally present across breeds and age classes [25]. At the same time, we know about only a very limited number of attempts where the researchers would focus on the intrinsic factors that could affect the dogs’ ability to learn via observing human demonstrators. 

So far probably, the only empirical attempts to connect social rank with non-competitive problem-solving behaviour in dogs was done by Pongrácz and colleagues [26,27]. In other species, it is widely established that rank and competitiveness show strong associations with the problem-solving ability of the individual. For example, in great tit males (*Parus major*), it was found that meanwhile the agonistic (competitive) behaviours show positive association with the performance in exploration, the less competitive (lower ranking) birds excel more in problem-solving [28]. In case of dogs, individual subjects of multi-dog households were assessed first by asking the owner to respond to four questions (barks first, eats first, wins fight, and its mouth is licked by other dog), whether they were dominant or subordinate compared to the “other” dog in the household [26]. Both dominant and subordinate dogs were tested in a detour test, without and with demonstration (i.e., to see their performance both in individual and social learning problem-solving situations). Trial and error success rate was similar (and poor) in both groups, but subordinates learned better from a dog, meanwhile dominants learned better from a human demonstrator. Dominant dogs did not learn at all from the other (unfamiliar) dog, but subordinates showed considerable learning capacity from the unfamiliar human demonstrator. “Singleton” dogs were not assessed as being dominant or subordinate, but were tested in the social learning groups and performed with a medium success. In another study, these results were confirmed in a different social learning scenario, where instead of the detour paradigm, subjects of multi-dog households were tested in an object manipulation (“tilting tube”) task with and without human demonstrator [27]. Dominant dogs performed better than subordinates in the trials after observing the demonstrator, but again, no rank-dependent difference was found in the individual problem-solving ability of the subjects.

Although the manifestation and sometimes even the existence of social rank systems are heavily debated in the framework of companion dogs [29,30], the biological relevance of inter-individual relationships that affect the dogs’ access and monopolizing potential of relevant resources is hardly questionable [31]. There is ample evidence that social dominance in dogs is associated with personality traits such as impulsivity, assertiveness, trainability and amicability [32,33,34], and these traits have an impact on behaviours in everyday life including owner-directed and problem behaviours, thus ultimately even on the quality of dog–owner relationship. As it can be assumed that the presence and attention of the owner might be one of the most valuable resources for a dog [35,36,37,38], therefore, it may also generate competition among dogs. So far, we know about one study where social rank of the dogs were tested among the potential factors that could influence the interactions between dogs in the owner’s presence as well as owner-directed behaviours in the presence of the other dogs (e.g., jealousy), although with inconclusive result [35].

The social rank of dogs seems to be associated with the quality and efficacy of a wide array of social interactions, including the dog–dog [26] and dog–human dimensions of social learning [26,27] as well as the broader aspect of personality traits and various behavioural problems [32]. However, it is not known so far, whether dogs’ performance in individual and social problem-solving tasks (i.e., trial-and-error and social learning) would be also connected to their individual-specific behavioural patterns during their interactions with their human partners. In other words, if we position social rank, social learning and dog–human relationship to each corner of a triangle, the empirical evidence is still missing that there would be an association between the latter two. The aim of this study was to investigate how certain aspects of the dog–owner relationship associate with social learning from familiar and unfamiliar humans. We hypothesized that the quality of the relationship (including aggressive tendencies, obedience and training techniques) will have an effect on performance in a social learning task. Specifically, we predicted that dogs that show a more harmonious relationship with their owners (higher obedience and lower aggression) would show better performance in a social learning situation. We also predicted that positive training techniques, especially those that are built on the development of intense attention and willingness to interact with the owner, would associate with better social learning performance. Based on our earlier results [39], we did not expect any difference in the efficacy of social learning in case of a familiar (the owner) or an unfamiliar (the experimenter) demonstrator. We predicted no effect of the dog–owner relationship on performance in an individual problem-solving situation.

## 2. Materials and Methods

### 2.1. Ethical Statement

Our study was conducted at Eötvös Loránd University in Budapest, Hungary, where animal experiments are overseen by the University Institutional Animal Care and Use Committee (UIACUC). According to the definition of “animal experiments” by the currently operating Hungarian law—the Animal Protection Act—our non-invasive observational experiment on dog behaviour was not considered as an animal experiment under the law and was therefore allowed to be conducted without any special permission from the UIACUC. All participation by owners with their dogs was voluntary, and owners stayed with their dog during the study. The owners filled out a consent form to permit their dogs to participate in the study, and to allow use of the resulting media in publications.

### 2.2. Subjects

Subjects of the study were companion dogs above 10 months (N = 98, M_age_ = 4.1 years; SD = 3.4; 52 female, 27 spayed; 46 male, 13 neutered; 85.7% of owners were female, 14.3 male, Mage = 30 years, SD = 1.85), recruited from attendees of various dog training facilities in Hungary. Participation was voluntary and no incentive was offered.

The tests were performed in secluded outdoor areas of the dog schools, familiar to the owners and dogs. Besides the owners and dogs, a female experimenter (G.R.) was present.

### 2.3. The Questionnaire

The owners were first asked to fill out a questionnaire consisting of 20 items about their dogs’ obedience and aggressive tendencies [40]. They had to mark their answers on a 10-point Likert scale (Table 1). They completed the questionnaires before the tests so their dogs’ behaviour and performance during the tests would not affect their responses later. Similarly, the experimenter did not evaluate the questionnaires on site to avoid forming expectations about the dogs’ behaviour.

### 2.4. Detour

In this test, the dogs had to detour around a wire mesh fence to obtain a reward (their favourite treat or favourite toy) [26,39]. The dogs were grouped in three groups: the Control group (N = 36), the Experimenter demonstration (N = 31) and the Owner demonstration group (N = 31). The test consisted of three trials, each lasted for maximum 60 s. The experimental setup can be seen on Figure 1.

In the Control condition, the owner stayed with the dog at the starting point (restraining the dog by holding its collar), the experimenter held the reward in her hand, called the dog’s attention with ostensive cues (calling the dog’s name, saying “Look!” in Hungarian) and showed him/her the reward. The experimenter then walked to the corner of the fence in a straight line and placed the reward in the inner corner from above (leaning over the top edge of the fence), showed her empty hands to the dog, then returned in a straight line. The owner let the dog free and the 60 s trial started. The owner was not allowed to give commands or point at the ends of the fence but could encourage the dog to keep going. The trial ended when the dog reached the reward (by detouring the fence) or the 60 s ended. Then, two similar trials followed.

The first trial of both demonstration groups was the same as for the control group. In the second and third trials in the Experimenter demonstration group the experimenter did not place the reward in the inner corner from above but walked around the fence and put down the reward in the corner, and left it there by walking back along the other side of the fence. If the dog was successful in the first trial, the experimenter detoured around the fence from the other side than the dog did.

In the second and third trials in the Owner demonstration group, the experimenter kept the dog at the start point while the owner carried the reward to the inner corner of the fence in a similar way as it was described in case of the Experimenter demonstration group, then the owner returned to the dog and he/she was the one to let the dog free.

The tests were recorded by a video camera and later coded with Solomon coder (beta 19.08.02 ^®^ András Péter, https://solomon.andraspeter.com, accessed on 23 February 2021).

### 2.5. Take-Away-Bone and Roll-Over Tests

As consecutive tests of potentially aggressive responses might increase the animal’s aggression [40,41], there was a 2–3 min break between these two tests to minimize the risk of this effect.

In the take-away-bone test, the dog was tethered to a tree or a fence post with a 2-m-long rope. The owner gave a large bone to the dog. There was a string attached to the bone, the owner held the other end of the string. In the other hand of the owner was an artificial hand. When the dog chewed the bone for 30 s, the owner stroked the dog’s back gently with the artificial hand, then asked for the bone. If the dog let go of the bone, the owner pulled it with the string and the test was over. If not, the owner placed the artificial hand on the bone for 5 s. If the dog still would not let go of the bone, the owner tried to pull the bone away with the string (while keeping the hand on the bone). The test was over once the dog let go of the bone or tried to attack the hand.

In the roll-over test, the dog was again tethered to the tree or fence post and muzzled. As wearing a muzzle is mandatory for dogs on all public transportation in Hungary, the dogs were familiar with it. The owner tried to gently make the dog lay down on its back without giving a command, only physically moving the dog but at the same time, not forcing it. The owner tried to keep the dog in this position for 30 s. If the dog stood up and the owner had to reposition it, the 30 s restarted. The test lasted for maximum 2 min from the onset of the owner’s first attempt to roll over the dog.

### 2.6. Statistical Analyses

We used R statistical software (R Development Core Team, 2015) in RStudio (RStudio Team, Boston, MA, USA) with packages psych, paran, caret, coxme, survival, survminer, emmeans, MuMIn and dplyr.

In the take-away-bone and roll-over tests, we scored how easily the owner can perform the task (how easily the dog lets go of the bone and how easily it can be rolled over) (see Table 2).

We performed PCA on the questionnaire to reduce the number of variables. First, we excluded items with near zero variance, then checked the correlation matrix for near perfect correlations but found none. We also used Kaiser–Meyer–Olkin Measure of Sample Adequacy. We ran parallel analysis to determine the number of the extracted components, then ran the Principal Component Analysis (with oblimin rotation). We excluded parameters with low (less than 0.5) or double loadings. Cronbach’s α was calculated to assess internal consistency.

In the detour test, we measured the latency to obtain the reward in each trial. Time was measured between the dog’s collar was released by the owner until the dog detoured the fence and touched the reward. When dogs could not solve the detour problem, a flat 60 s latency was assigned to the trial. Additionally, we measured also the frequency the dog looked back at the owner during the trial (number of look-back divided by the latency).

We used Cox Regression Models to analyse the latency in the detour test, with subjects as random factor. We tested the association with the scores in the Take-away-bone and Roll-over tests, the scores for the principal components of the questionnaire and the regularity of the use of training method “shaping” (a specific training method in clicker training used to teach complex behaviour sequences by reinforcing simple behaviour building blocks the animal voluntarily offers). We used bottom-up model selection to find the most parsimonious models. We used GLM to see how looking back at the owner for help associated with the fixed factors. We applied AIC based model selection to find the parsimonious models.

## 3. Results

The item “rest_growl” (“If being disturbed while resting, the dog growls or snaps.”) had near-zero variance, so we removed it from the dataset before performing the PCA. KMO revealed an MSA of 0.66. The PCA on the questionnaire items revealed three principal components based on a total of 11 items, which were labelled “overactive,”, “irritable” and “attachment.” After examining the internal consistency of the components, we found “attachment” to be inconsistent (Cronbach’s α = 0.44), so we only used “overactive” and “irritable” in the further analyses (Table 3).

### 3.1. Latencies

In the Control group (no demonstration), we found a non-significant trend (*p* = 0.056) with the number of trials that suggests that trial-and-error learning may have a weak effect in learning the detour task (Figure 2A). We found no association between the latency and any other variable in the control group.

In the Experimenter demonstration group, the repetition of trials had a significant effect, thus indicating the effect of demonstration. Tukey post hoc tests showed that dogs became significantly faster in trials 2 and 3 compared to trial 1 (95% CI = (0.239–0.745), Z = 2.93, *p* = 0.0096) and there was no significant difference between the second and the third trial (Figure 2B).

More irritable dogs were also faster to solve the detour in this group (95% CI = (1.161–1.399), Z = 5.11, *p* < 0.0001). This effect of the factor “irritable” was only significant after the demonstration (in trials 2 and 3) (Figure 3).

We also found a significant association with the variable “shaping.” The more a dog was trained with the method of shaping, the faster it solved the detour task (95% CI = (1.054–1.260), Z = 3.1, *p* = 0.0019). Behaviour in the Take-away-bone test also associated with the dogs’ latencies in the detour test: post-hoc tests showed that dogs who immediately let go of the bone solved the task faster than dogs that only gave it up when the artificial hand rested on it (95% CI = (1.782–12.604), Z = −3.12, *p* = 0.0156) or only after some tugging (95% CI = (1.853–10.646), Z = −3.35, *p* = 0.0074), and this association was only significant after the demonstration (in trials 2 and 3) (Figure 4).

In the Owner demonstration group, only the effect of the demonstration was significant: post hoc tests showed that dogs solved the detour faster after demonstration (trials 2 and 3) than in trial 1 (95% CI = (0.193–0.616), Z = 3.59, *p* = 0.001) (Figure 2C).

### 3.2. Looking Back at the Owner

In the Control group, none of the fixed factors had a significant effect on looking back at the owner.

In the first trial (without demonstration) in both the Experimenter demonstration and Owner demonstration groups, we found that those dogs that scored higher on “overactive” looked back less at the owner (Experimenter demonstration: 95% CI = (1.683–9.323), Z = 2.75, *p* = 0.0102, Owner demonstration: 95% CI = (2.593–8.913), Z = 3.49, *p* = 0.00156) (Figure 5).

In trials 2 and 3 (after the demonstration), the only variable that showed a significant association with the frequency of looking back was “irritable” in both groups: more irritable dogs looked back at the owner less frequently (Experimenter demonstration: 95% CI = (0.606–6.032), Z = 2.46, *p* = 0.0169, Owner demonstration: 95% CI = (0.160–3.636), Z = 2.1, *p* = 0.04) (Figure 6).

The final models for both the latencies and frequencies of looking back at the owner are shown in Table 4.

## 4. Discussion

In this experiment, we successfully replicated the earlier results of Pongrácz et al. (2001, 2003, 2004), showing that dogs learn the detour task more quickly and more efficiently when observing a human demonstrator, than via trial-and-error learning. We also confirmed that similarly to Pongrácz et al. [39] the familiarity with the demonstrator (owner vs. unfamiliar experimenter), in general, does not affect the social learning performance of the dogs. However, in line with our predictions, the individual dogs’ social behavioural traits that characterize its relationship and everyday interactions with the owner, as well as the dog’s behavioural responses in some slightly confrontational contexts with the owner (“Take away bone”) did show significant associations with the problem-solving performance in the detour test. Importantly, these effects always manifested themselves in the social learning context, while the trial-and-error attempts of the dogs remained unaffected by these factors. Another remarkable finding was that the abovementioned associations between particular traits of the dogs (related to the social relationship with the owner, such as “irritable”) and the social learning performance arose only in case of the demonstration by the experimenter, meanwhile, after observing their owner as demonstrator, dogs’ performance did not show significant association with their behavioural scores from the questionnaire and the behavioural tests.

It was found in several studies that dogs look (back) at the nearby owner in such situations when they face a difficult/unsolvable problem [42], or an unusual/frightening stimulus [43]. In earlier experiments with the detour paradigm (e.g., [39]), high frequencies of looking back at the owner correlated with the difficulty of the task, i.e., in other words, dogs looked back at the owner more frequently when they showed a less successful detour performance. In our current study, we found similar results, because dogs with higher scores on the “irritable” factor performed better in the detour task and also looked back at the owner less frequently in case of the trials with human demonstration.

According to the significant negative association, dogs that scored high on the “overactive” factor, looked back less frequently at the owner in both experimental groups. However, this was true only in the individual problem-solving trials. This suggests that the items belonging to this factor might not relate to the dogs’ social behaviour towards the owner. This result shows an interesting parallel with the results of Bunford et al. [44], who found that in a Go/no-Go task, those dogs were more prone to the so-called commission error, who scored high on the inattentiveness/impulsivity and hyperactivity scales. The commission error represents a behaviour when the subject responds when it should not, in other words, when it shows a more persistent, less inhibited response. In our test, when they were left on their own, the more “overactive” subjects showed less dependent behaviours (looking back) and may tried to solve the problem more persistently compared to the less “overactive” (or less inattentive) dogs, which shows a functional similarity to the inattentive/hyperactive dogs in the Bunford et al. study.

The factor “irritable” had the strongest association with the effect of social learning from an unfamiliar demonstrator in the detour task. According to the owners, dogs with high “irritable” scores may behave assertively in everyday situations and do not tolerate well such interactions with humans that may cause certain levels of stress or uncomfortable feelings. However, according to the results of our experiment, these dogs utilized much more effectively the unfamiliar human’s detour demonstrations than the dogs with low scores on the “irritable” factor. The only other comparable result to this was found by Pongrácz et al. [26], where high-ranking (“dominant”) dogs showed more effective social learning from the human demonstrator than the low-ranking (“subordinate”) dogs did. As in the present study, the hierarchy status of the subjects was not evaluated, we cannot tell whether the highly “irritable” dogs are more likely the high ranking ones, too. Rank positions are only meaningful if there are multiple dogs in the household, and in this study singleton and multi-dog household subjects were equally tested.

Based on the results, we can assume that the highly “irritable” dogs may be more attentive to the human actions, and this could cause that they paid more keen attention to the human demonstrator, thus could learn more effectively from her behaviour. An interesting parallel with this explanation was found in semi-captive working Asian elephants, where researchers detected a rather strong positive correlation between the “aggressiveness” and “attentiveness” of these highly social and intelligent animals [44]. In that paper, the items that belong to the elephants’ aggressiveness factor were somewhat similar to the items in case of the factor “irritable” in dogs, and “attentiveness” in the elephants contains items that would likely enhance social learning performance in dogs as well. The importance of dogs’ attention in the most crucial moments of human demonstration was earlier confirmed by Péter et al. [45]. In that study, dogs could successfully find a hidden object after a video-demonstration only if they paid attention to the human demonstrator in the very moment when he/she performed the hiding movement with the target. However, then why this association between the “irritable” scores and social learning performance was not found in case of the owner-demonstration trials? If we consider the owner as the most fundamental source of reference for the dogs [43,46,47], one could assume that dogs pay a uniformly high level of attention to the detour action of the owner, independently of their general attitude with human interactions. 

The factor “irritable” consists of items describing aggression-related behaviours but also assertiveness and persistence. Previous studies have found that dominant dogs from multi-dog households learn more effectively from unfamiliar human demonstrators than the subordinate dogs do [27], and interestingly, the factor “irritable” had a similar association with the detour-performance in our study. The items building up the “irritable” factor are the ones that describe certain aspects of the social behaviour of dogs (e.g., growling, biting, and behaving in an assertive way). The other important factor associated with social learning performance was possessiveness that we measured with the “Take-away-bone” test, which was earlier found to be associated with aggressive tendencies [40], but here we found opposite association with social learning. We also found that characteristics that are less social in nature (e.g., inattention and inhibition) do not affect social learning performance and that the examined social characteristics do not affect individual problem-solving at all.

Obviously, the found associations among some factors describing the dog–owner relationship and the dogs’ social learning performance do not refer to any causative relationship between the variables. Our study, therefore, can be regarded as “correlative”–while in the future, by relating the performance in the social learning test with a longitudinal behavioural assessment of dogs that undergo a training procedure aimed at behavioural problems affecting the “irritable” component could shed more light on the causal connections between dog–human relationship and social learning.

## 5. Conclusions

The capacity for social learning from a human demonstrator is a robust phenomenon in companion dogs, which so far was found to be only associated with the social rank of a given dog from multi-dog households (ref). In the present study, we outlined a new paradigm, showing that some factors that characterize the dog–owner relationship (tendency to be aggressive with the owner or being in general irritable in particular contexts) may also be associated with such social behaviours, as the capacity from learning via observing a human demonstrator. Our results call the attention to the complexity of dog–human interactions, as well as open up new, promising possibilities to the better understanding of the interplay between dog–dog social dynamics and the relationship between companion dogs and their owners.

## Figures and Tables

**Figure 1 animals-11-00961-f001:**
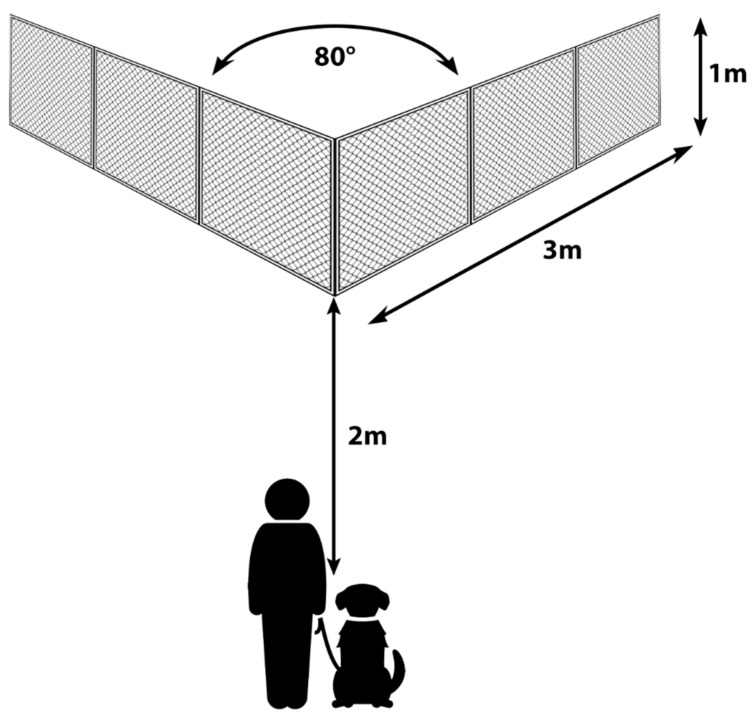
The experimental setup of the detour test.

**Figure 2 animals-11-00961-f002:**
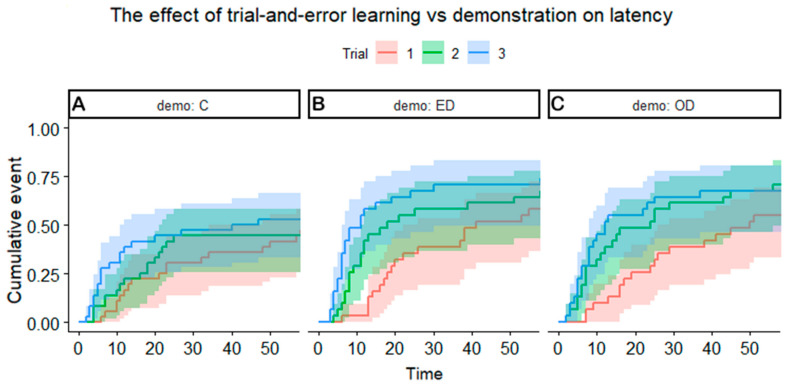
(**A**) Trial-and-error learning had a weak effect on learning the detour task. Dogs solved the detour faster after the demonstration both when (**B**) the experimenter demonstrated the task and (**C**) the owner demonstrated the task. There was no significant difference in the latency between the second and third trials. Abbreviations: demo: C = control; demo: ED = experimenter demonstration; demo: OD = owner demonstration.

**Figure 3 animals-11-00961-f003:**
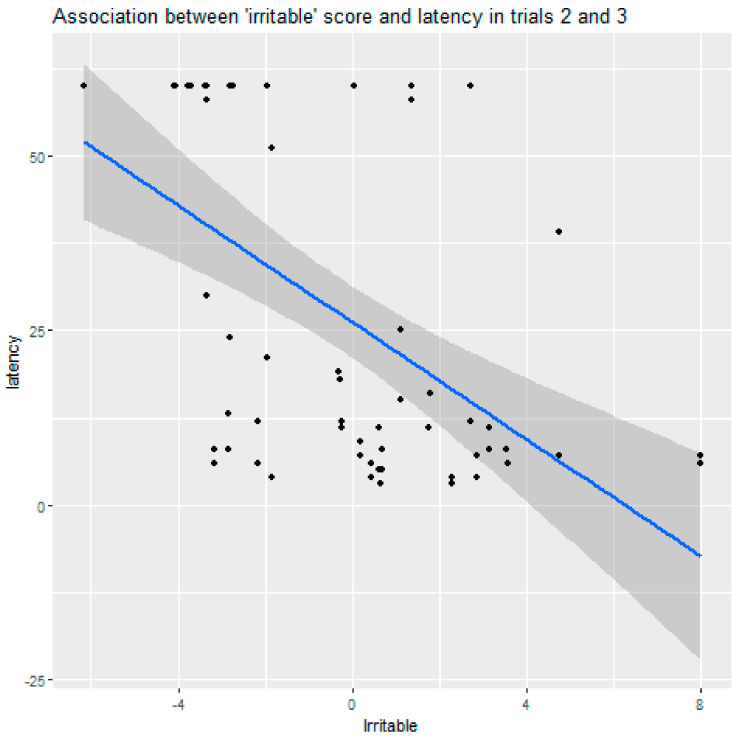
More “irritable” dogs solved the detour faster after the experimenter demonstrated the task.

**Figure 4 animals-11-00961-f004:**
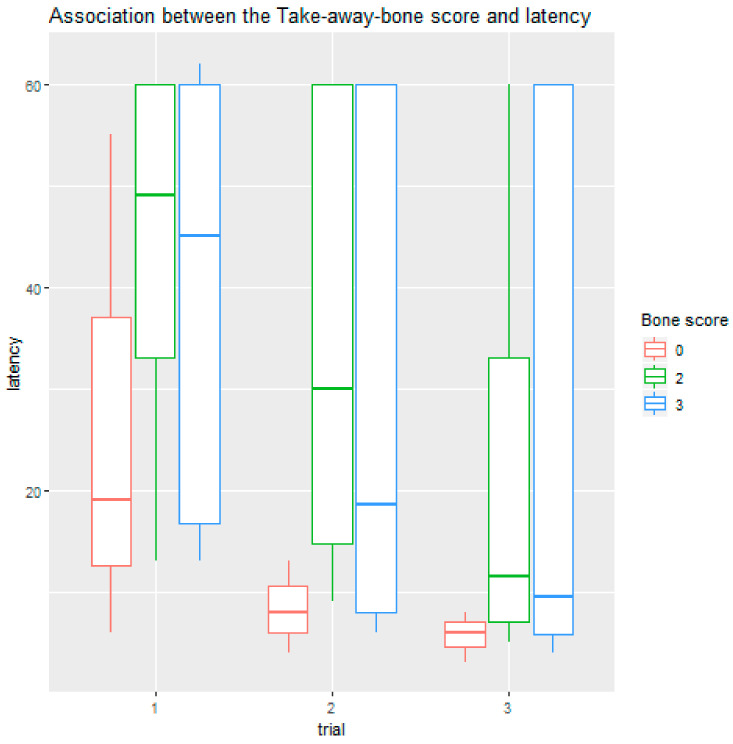
Dogs that gave the bone up the easiest in the Take-away-bone test solved the detour faster after the experimenter demonstrated the task.

**Figure 5 animals-11-00961-f005:**
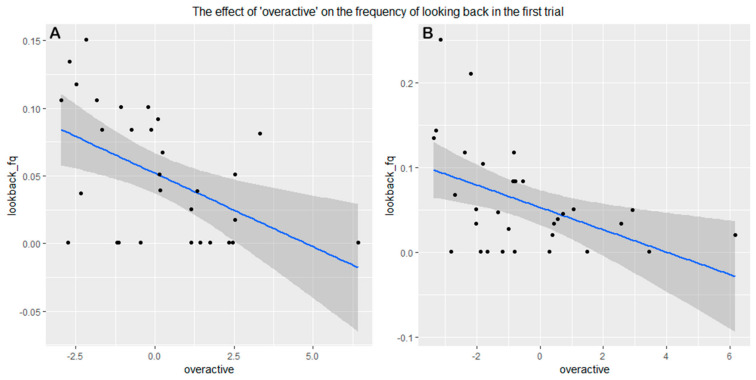
More “Overactive” dogs looked back less frequently at the owner in the first trial in both the (**A**) experimenter demonstration group and (**B**) owner demonstration group.

**Figure 6 animals-11-00961-f006:**
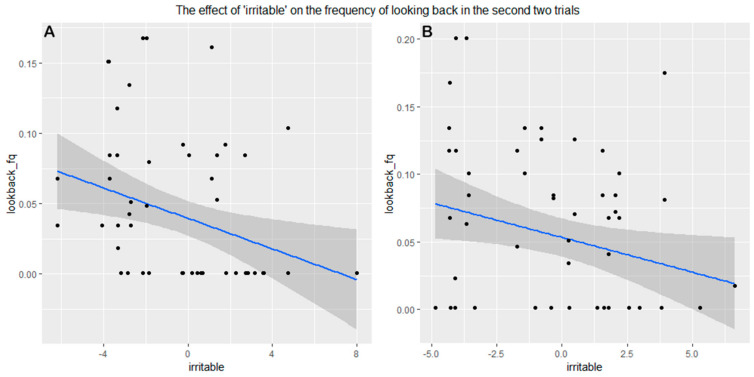
More “irritable” dogs looked back less frequently at the owner in the second and third trials in both the (**A**) experimenter demonstration group and (**B**) owner demonstration group.

**Table 1 animals-11-00961-t001:** The questionnaire used in the study.

The 20 Questions Used to Evaluate Dog–Owner Relationship and Problematic Behaviours.	Variable Name
The dog can be called back even if there are other dogs, animals or humans in its vicinity.	callback
The owner can easily stop unwanted activities (e.g., by verbal inhibition).	inhib
Sometimes, the dog becomes so overactive during play that the activity has to be ended.	overexc
The dog intensely defends its food, ball or other assets, even from the owner.	resguard
The dog has a skill to seek out and steal food from anywhere, sometimes even from the hands of people.	steal
The dog demands physical contact with the owner: it often cuddles or snuggles up to the owner or leans its head in the owner’s lap.	contact
The dog growls when being groomed, bathed or when the paws/ears are being cleaned.	groom_growl
If being disturbed while resting, the dog growls or snaps.	rest_growl
The dog seizes every opportunity to escape and run away, and after successfully getting away, it is very difficult to call him back.	escape
The dog follows the owner whenever and wherever it is possible.	follow
The dog might bite or snap at others (humans or dogs) in the presence of the owner.	bite
The dog responds by barking or growling to situations/events it does not appreciate or opposes.	growl
The dog responds threateningly/shows intimidating behaviour if being punished or disciplined.	talks_back
The dog is highly frustrated when left alone, continuously barks or shows destructive behaviour.	separation
If the dog wants to obtain something, it pursues that persistently or even aggressively.	pursue
The dog behaves in an assertive manner.	assertive
If the dog once understands that something is forbidden, it is easy to prevent the same thing on a subsequent occasion.	easy_forbid
Sometimes, the dog’s attention is so distracted that it impairs its obedience.	inattention
The dog often barks in unusual or novel situations. In these cases, it is almost impossible to calm it.	bark
During clicker training, the dog is usually trained by the so-called shaping method.	shaping

**Table 2 animals-11-00961-t002:** Variables measured in the Take-away-bone and Roll-over tests based on Bálint et al. [40].

Behaviour	Score
Take-away-bone	
The dog releases the bone when its back is being stroked with the artificial hand	0
The dog releases the bone when the owner asks for it or reaches for it with the artificial hand	1
The dog releases the bone when the artificial hand rests on it	2
The dog releases the bone after some tugging	3
The dog does not release the bone	4
Roll-over	
The dog does not show any resistance	0
The dog resists once, but then can easily be laid on its back	1
The dog resists/gets up more than once, but eventually can be laid on its back	2
The dog resists throughout the whole test	3

**Table 3 animals-11-00961-t003:** The three principal components revealed by PCA on the questionnaire items, and the loadings of each item. Note: principal component “attachment” was eventually excluded from further analysis because of its weak internal consistency.

	Overactive	Irritable	Attachment	
Callback	0.852852			The dog can be called back even if there are other dogs, animals or humans in its vicinity.
Inattention	0.773509			Sometimes, the dog’s attention is so distracted that it impairs its obedience.
Inhib	0.723818			The owner can easily stop unwanted activities (e.g., by verbal inhibition).
Escape	0.682582			The dog seizes every opportunity to escape and run away, and after successfully getting away, it is very difficult to call it back.
Growl		0.761661		The dog responds by barking or growling to situations/events it does not appreciate or opposes.
Pursue		0.553041		If the dog wants to obtain something, it pursues that persistently or even aggressively.
Assertive		0.538901		The dog behaves in an assertive manner.
Bite		0.525492		The dog might bite or snap at others (humans or dogs) in the presence of the owner.
Groom_growl		0.501269		The dog growls when being groomed, bathed or when its paws/ears are being cleaned.
Contact			0.771242	The dog demands physical contact with the owner: it often cuddles or snuggles up to the owner or leans its head in the owner’s lap.
Separation			0.655851	The dog is highly frustrated when left alone, continuously barks or shows destructive behaviour.

**Table 4 animals-11-00961-t004:** The final models as reported in the results.

**Models for Latencies (Cox Regression Model)**					
Experimenter demonstration group		Est.	Standard Error	Z	*p*
	Trial				Main effect *p*: < 0.0001 ***
	Trial 1–2	0.857	0.0293	2.927	0.0096 **
	Trial 1–3	1.399	0.313	4.469	<0.0001 ***
	Irritable	0.242	0.0475	5.11	<0.0001 ***
	Shaping	0.142	0.0457	3.1	0.0019 **
	Bone				Main effect *p*: 0.03412 *
	Bone 0–2	−1.556	0.499	−3.12	0.0156 *
	Bone 0–3	−1.491	0.446	−3.345	0.0074 **
Owner demonstration group		Est.	Standard Error	Z	*p*
	Trial				Main effect *p*: 0.0002 ***
	Trial 1–2	1.064	0.296	3.594	0.001 **
	Trial 2–3	1.186	0.268	3.881	0.0003 ***
**Models for looking back to the owner (GLM)**					
First trial (without demonstration)					
Experimenter demonstration group		Est.	Standard Error	t	*p*
	Overactive	5.326	1.939	2.746	0.0102 *
Owner demonstration group		Est.	Standard Error	t	*p*
	Overactive	5.621	1.610	3.492	0.00156 *
Second and third trials (after demonstration)					
Experimenter demonstration group		Est.	Standard Error	t	*p*
	Irritable	3.439	1.399	2.458	0.0169 *
Owner demonstration group		Est.	Standard Error	t	*p*
	Irritable	1.8483	0.8802	2.1	0.04 *

*p* *: < 0.05; **: < 0.01; ***: < 0.001.

## Data Availability

The data presented in this study are available on request from the corresponding author. The data are not publicly available due to the reason that our subjects were privately owned companion dogs.

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
