# Peer review of "Grumpy Dogs Are Smart Learners—The Association between Dog–Owner Relationship and Dogs’ Performance in a Social Learning Task"

_animals, 2021, doi:10.3390/ani11040961_

Round 1
Reviewer 1 Report
The authors used a combination of owner questionnaire and two behavior tests (possession of bone and roll-over) to describe owner-dog relationships and examined how these relationships affected social learning (assessed via a detour test). The sample size was large (n = 98 dogs), and the findings are interesting. My specific comments are listed below.
General comment: The actual order of information gathering/testing was (1) questionnaire, (2) social learning task, and (3) behavior tests (possession of bone and roll-over), which makes sense given the potentially confrontational nature of the two behavior tests. However, throughout the manuscript (e.g., Abstract, Simple Summary, Materials and Methods) the coverage occurs in the following order: questionnaire, behavior tests (possession of bone and roll-over), and social learning task. I understand that the questionnaire and behavior tests both assess the owner-dog relationship, which is then examined for its effects on social learning, but the coverage in the text is very confusing. For example, lines 11-12, “We used a questionnaire and two behavior tests to assess the dogs’ attitude towards the owner, then tested the dogs in a well-established detour reaching task…” – this is misleading and confusing because it is not the order in which tests actually occurred. Please clarify this issue throughout; the comment on lines 173-174 is insufficient.
Simple Summary and Abstract: Both would benefit by including a last sentence that puts the findings in a larger context. As currently written, both end with results.
Introduction:
Lines 50 to line 54: Please clarify this sentence.
Lines 118-125: The authors present their hypotheses, but do not provide specific predictions and reasons for predictions (e.g., What specific differences did you expect when the demonstrator was familiar versus unfamiliar and why?)
Material and Methods:
It would be good to include basic information on the owners, such as age and sex, and possibly also how long they had owned their dog.
Line 139: Include years as the unit of measure for age here.
Line 140: Readers may not know what is meant by “Hungarian dog schools”, so a brief description would be helpful.
Line 148: Is the questionnaire in the supplemental material for this paper? Readers should not have to go to another paper [39] to see the specific questions.
Line 176: Please provide the number of dogs in each group: Control, Owner demonstration, and Experimenter demonstration.
Line 184: Give an example or two of “ostensive cues”.
Line 228: This is the first time the term “shaping” appears (this is why the questionnaire should be available in the paper); please describe this method of training for readers since it is one of the variables analyzed.
Results:
What do the numbers in Table 2 represent? This should be made clear either in the title of the table or a footnote.
Lines 240-243: The P value of 0.056 is correctly described as a trend, but should not be dismissed. It would be good to show these results in graph form for completeness.
Discussion:
Lines 293-295: But the results described in lines 240-243 approach significance (P value of 0.056), so this statement here is not strictly correct. Maybe rephrase this using something like: “dogs learn more quickly when observing a human demonstrator, than via trial-and-error learning”?
The Discussion should include coverage of study limitations.
Finally, I believe this journal requires a Conclusions section, which is missing from the manuscript. Also missing is some of the backmatter material, at least in the version I downloaded (e.g., Author contributions, Acknowledgments).
Author Response
RESPONSES TO REVIEWER 1
First of all, we are really thankful for all the supportive comments and questions. By answering these, we believe that we could significantly improve our manuscript
General comment: The actual order of information gathering/testing was (1) questionnaire, (2) social learning task, and (3) behavior tests (possession of bone and roll-over), which makes sense given the potentially confrontational nature of the two behavior tests. However, throughout the manuscript (e.g., Abstract, Simple Summary, Materials and Methods) the coverage occurs in the following order: questionnaire, behavior tests (possession of bone and roll-over), and social learning task. I understand that the questionnaire and behavior tests both assess the owner-dog relationship, which is then examined for its effects on social learning, but the coverage in the text is very confusing. For example, lines 11-12, “We used a questionnaire and two behavior tests to assess the dogs’ attitude towards the owner, then tested the dogs in a well-established detour reaching task…” – this is misleading and confusing because it is not the order in which tests actually occurred. Please clarify this issue throughout; the comment on lines 173-174 is insufficient.
RESPONSE: Thank you for this comment, the Reviewer is obviously right. As the Reviewer mentioned, because both behaviour tests plus the questionnaire were used to assess the dog-owner relationship, it was somewhat logical to mention them in a clustered manner – however we agree that this could be confusing for the reader regarding the actual order of the various tests and assessments. Therefore we used a more careful wording now throughout the manuscript, and put the description of the individual tests and assessments in that order as they actually happened to the Methods section.
Simple Summary and Abstract: Both would benefit by including a last sentence that puts the findings in a larger context. As currently written, both end with results.
RESPONSE: Unfortunately, the summary section is strongly limited (200 words), but we agree with this comment of the Reviewer and added a sentence about the possible future research directions that can be based on present findings.
Introduction:
Lines 50 to line 54: Please clarify this sentence.
RESPONSE: We elaborated this section a little more, providing direct information about the crucial details of the cited study.
Lines 118-125: The authors present their hypotheses, but do not provide specific predictions and reasons for predictions (e.g., What specific differences did you expect when the demonstrator was familiar versus unfamiliar and why?)
RESPONSE: Thank you for the request, now we elaborated this section with adding specific predictions to the hypotheses.
Material and Methods:
It would be good to include basic information on the owners, such as age and sex, and possibly also how long they had owned their dog
RESPONSE: Thank you for the suggestion, basic owner demographics were included to the revised version.
Line 139: Include years as the unit of measure for age here.
RESPONSE: Done.
Line 140: Readers may not know what is meant by “Hungarian dog schools”, so a brief description would be helpful.
RESPONSE: We rewrote this section, emphasizing that the subjects were recruited with their owners from dog schools in Hungary.
Line 148: Is the questionnaire in the supplemental material for this paper? Readers should not have to go to another paper [39] to see the specific questions.
RESPONSE: Now there is a new table (Table 1) in the manuscript that contains each of the 20 questions that were used to evaluate the dog-owner relationship.
Line 176: Please provide the number of dogs in each group: Control, Owner demonstration, and Experimenter demonstration.
RESPONSE: The requested details were added to the manuscript.
Line 184: Give an example or two of “ostensive cues”.
RESPONSE: Done.
Line 228: This is the first time the term “shaping” appears (this is why the questionnaire should be available in the paper); please describe this method of training for readers since it is one of the variables analyzed.
RESPONSE: Thank you for noticing this, now we elaborated the description of this method.
Results:
What do the numbers in Table 2 represent? This should be made clear either in the title of the table or a footnote.
RESPONSE: As we added an extra Table to the manuscript, the table in question is now became Table 3. We explain in the footnote that the numbers are the so-called ‘loadings’, which represent the importance of an item in a given principal component.
Lines 240-243: The P value of 0.056 is correctly described as a trend, but should not be dismissed. It would be good to show these results in graph form for completeness.
RESPONSE: Complying with this request, we added the new Figure. Originally we avoided this as it happens often that reviewers argue against the overrepresentation of trend-like results in the papers.
Discussion:
Lines 293-295: But the results described in lines 240-243 approach significance (P value of 0.056), so this statement here is not strictly correct. Maybe rephrase this using something like: “dogs learn more quickly when observing a human demonstrator, than via trial-and-error learning”?
RESPONSE: Thank you for this thoughtful suggestion, the discussion was amended accordingly.
The Discussion should include coverage of study limitations.
RESPONSE: Thank you for this important note, we added this text towards the end of the Discussion: “Obviously, the found associations among some factors describing the dog-owner relationship and the dogs’ social learning performance do not refer to any causative relationship between the variables. Our study therefore can be regarded as ’correlative’ – whilst in the future, by relating the performance in the social learning test with a longitudinal behavioural assessment of dogs that undergo a training procedure aimed at behavioural problems affecting the ’irritable’ component could shed more light on the causal connections between dog-human relationship and social learning.”
Finally, I believe this journal requires a Conclusions section, which is missing from the manuscript. Also missing is some of the backmatter material, at least in the version I downloaded (e.g., Author contributions, Acknowledgments).
RESPONSE: The Reviewer is right – therefore we added a Conclusion paragraph. We are not sure whether Author contributions /Acknowledgements are available for the reviewers during the peer-review process, we can confirm here that we supplied these during the manuscript submission.
Reviewer 2 Report
Reviewer comments
Manuscript ID: animals-1142927
Title: Grumpy dogs are smart learners – the association between dog-owner relationship and dogs’ performance in a social learning task
Authors: Péter Pongrácz *, Gabriella Rieger, Kata Vékony
This is an interesting study on observational/social learning on dogs in relationship with a human demonstrator investigating the role of dog-owner relationship.
The paper is in line with the objectives of the journal “Animals”. I think that this research is timely and relevant because the authors present data that add new knowledge to the matter and is important from a practical point of view. However, there are some indications that require some specification. Particularly, the authors used the term “social” learning extensively in the text but in lines 50 and 305 they use “observational” learning. The two mechanisms are not identical (Zentall, 2006). Please explain.
Zentall, T. R. (2006). Imitation: definitions, evidence, and mechanisms. Animal cognition, 9(4), 335-353.
Looking back is an indication that was introduced in the impossible task paradigm for the position of the owner but here is misleading. I suggest using "gazing" instead of "looking back" in the whole text.
There are some little typing mistakes (see for example, Line 64, remove space between [21] and full stop, the same in line 140, 3. 4, and in other lines) but in general the paper will benefit from professional English editing.
Lines 110-112: Please provide a reference.
Line 124: the term “asocial” is inappropriate. Replace with “physical”. Please check and change also in other places in the text.
Line 127 (Ethical statement). My opinion is that dogs have been stressed by the take-away-bone and roll-over tests. It is unclear how triggering aggressive tendencies is a normal ethological test not affecting the welfare. Clarification is requested.
Line 148: the questionnaire should be given in this paper not remanding the reader to another paper.
Line 222-4: please explain how you interpreted the looking back (gazing) behavior when the owner was the demonstrator. It has a significance different from when the stranger is the demonstrator. In the latter, it could be a social referencing but not when following the dynamic of the owner during its demonstration.
Lines 233-7: the Kaiser-Meyer-Olkin (KMO) as a measure of how suited your data is for the Analysis should be given.
Lines 314-6: alternatively, they gaze less since they spend less time in complete the task.
Table 2: only one of the four parameters (i.e. escape) could be associated with the term “overactive”, but not match the other parameters. Moreover, the behaviors are in opposition. Particularly is not so easy to explain how obedience responses (callback and inhibition) and disobedient responses (inattention) correlated in the same principal component with positive loadings.
Author Response
RESPONSES TO REVIEWER 2
We would like to say “thank you” for all the constructive, useful comments and criticism on our manuscript.
The paper is in line with the objectives of the journal “Animals”. I think that this research is timely and relevant because the authors present data that add new knowledge to the matter and is important from a practical point of view. However, there are some indications that require some specification. Particularly, the authors used the term “social” learning extensively in the text but in lines 50 and 305 they use “observational” learning. The two mechanisms are not identical (Zentall, 2006). Please explain.
RESPONSE: Thank you for this note – we agree that social learning and observational learning are not fully synonymous terms. Now we uniformly use ‘social learning’ everywhere in the manuscript.
Looking back is an indication that was introduced in the impossible task paradigm for the position of the owner but here is misleading. I suggest using "gazing" instead of "looking back" in the whole text. –
RESPONSE: Actually, the “looking back” term as a behavioural variable has been already used in our first social learning paper on dogs (Pongrácz et al., 2001) – and just like there, we use it primarily as a reference to the actual topography of the specific gazing response of dogs in the detour scenario. What happens here, is that the dog leaves the owner at the start point and goes ahead to the fence, where it tries to detour the obstacle. When the dog looks at the owner during the one minute trial, this happens ‘backward’, because the owner stands behind the dog. Importantly, we coded this behaviour only during the actual trial period (from the dog’s departure from the start until it solved the detour or the trial elapsed without a successful detour). The Reviewer is right when writing that this behaviour is functionally identical with the ‘looking back’ response of dogs in case of the impossible task paradigm (e.g. Miklósi et al., 2003, Curr. Biol.). In our case, the specific topography of this behaviour (the dog actually looks back at the owner, who stands behind it) requires using it.
There are some little typing mistakes (see for example, Line 64, remove space between [21] and full stop, the same in line 140, 3. 4, and in other lines) but in general the paper will benefit from professional English editing.
RESPONSE: We corrected many smaller typos and the extra ‘spaces’ were deleted.
Lines 110-112: Please provide a reference.
RESPONSE: We added the requested references.
Line 124: the term “asocial” is inappropriate. Replace with “physical”. Please check and change also in other places in the text.
RESPONSE: We did it according to the suggestion.
Line 127 (Ethical statement). My opinion is that dogs have been stressed by the take-away-bone and roll-over tests. It is unclear how triggering aggressive tendencies is a normal ethological test not affecting the welfare. Clarification is requested.
RESPONSE: We agree that the behavioural tests might cause mild stress to some of the subjects, however, we carefully chose these methods as they were similar to such everyday situations that frequently occur in case of companion dogs. Rolling over the dog is a standard scenario at any veterinarian during a checkup, muzzling dogs is a standard procedure in Hungary at public areas, and we tested dogs with the take away the bone method where the owner indicated that this scenario is familiar to his/her dog. Additionally, if the owner felt the test causes stress to their dog, they could stop the test at any time. Therefore, we firmly believe that our experiments did not have any unwanted welfare consequences to the subjects.
Line 148: the questionnaire should be given in this paper not remanding the reader to another paper.
RESPONSE: We added a whole new Table with the 20 questions that we used for the description of dog-owner relationship.
Line 222-4: please explain how you interpreted the looking back (gazing) behavior when the owner was the demonstrator. It has a significance different from when the stranger is the demonstrator. In the latter, it could be a social referencing but not when following the dynamic of the owner during its demonstration. –
RESPONSE: We would like to clarify here again that ‘looking back’ at the owner is a behavioural variable that was only recorded during the actual trial, when the dog was attempting the detour. We did not record this behaviour during the demonstration phase. The owner was standing at the start point in every case when the dog was trying to detour the fence – therefore it is reasonable to call this behaviour as ‘looking back’, because dogs actually looked back at the owner behind them.
Lines 233-7: the Kaiser-Meyer-Olkin (KMO) as a measure of how suited your data is for the Analysis should be given.
RESPONSE: We added this analysis to the manuscript.
Lines 314-6: alternatively, they gaze less since they spend less time in complete the task.
RESPONSE: We used the frequency of looking back at the owner (occurrence of behavior divided by the length of the trial from the dog’s departure till it reaches the reward) to eliminate the variable’s sensitivity to the length of the trial. This way we could compare the looking back frequencies among trials of different lengths.
Table 2: only one of the four parameters (i.e. escape) could be associated with the term “overactive”, but not match the other parameters. Moreover, the behaviors are in opposition. Particularly is not so easy to explain how obedience responses (callback and inhibition) and disobedient responses (inattention) correlated in the same principal component with positive loadings.
RESPONSE: We found it a bit difficult to find fitting names to each principal component. We finally decided with ‘overactive’ because of the somewhat contradicting items it contained. We checked several times, and the loadings are indeed all positive. The reason we chose ‘overactive’ was the resemblance of this component to some ADHD symptoms.